# Prism-MoE: Efficient Dense-to-MoE Conversion for Visual Autoregressive Generation

**Ying Li** [* 1]  **Zefang Wang** [* 2]  **Zhaode Wang** [3]  **Zhiwen Chen** [3]  **Chengfei Lv** [3]  **Huan Wang** [1]

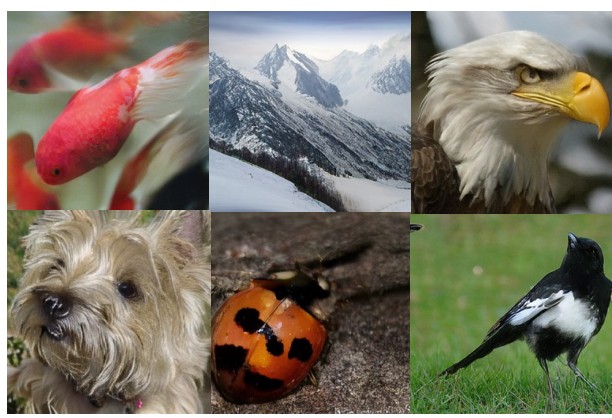 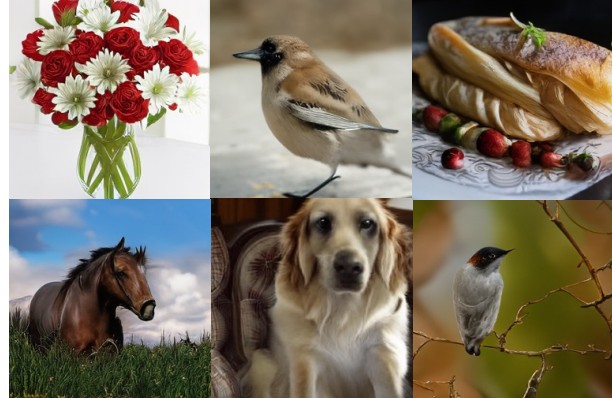

(a) VAR-d20, 37.5% active params, 8% training budget  (b) LlamaGen-XL Stage I, 37.5% active params, 6.7% training budget

*Figure 1.* **Prism-MoE** enables low-cost dense-to-MoE conversion with trajectory-aware initialization and condition-adaptive sparse fine-tuning, while maintaining competitive autoregressive image generation quality under sparse activation.

## Abstract

Scaling up visual autoregressive models improves generation quality but incurs substantial inference costs. Mixture-of-Experts (MoE) architectures mitigate this issue through sparse activation and have proven effective in large language models. However, training MoE models from scratch remains prohibitively expensive, and dense-to-MoE conversion for visual autoregressive models is still underexplored. To enable *low-cost and high-quality dense-to-MoE conversion*, we propose **Prism-MoE**, an efficient framework for transforming pretrained dense visual autoregressive models into sparse MoE models. Prism-MoE consists of two key components. First, we introduce trajectory-consistent initialization, which casts expert initialization as a structured optimization problem and preserves the generation trajectory of dense models. Second, we propose a confidence-adaptive sparse fine-tuning framework that aligns

expert specialization with the information density of visual tokens via confidence-aware routing supervision. Experiments show that Prism-MoE achieves dense-to-MoE conversion with less than **10%** of the standard training budget, while achieving generation quality close to dense baselines using only **37.5%** active parameters.

## 1. Introduction

Recent advances in autoregressive image generation (Tang et al., 2025; Sun et al., 2024; Fan et al., 2025; Chen et al., 2025) and multimodal modeling (Jin et al., 2025) have demonstrated strong performance alongside a clear trend toward increasing model capacity. This trend has driven rapid growth in parameter counts across architectures, from next-token generation models such as LlamaGen (Sun et al., 2024) and Janus-Pro (Chen et al., 2025) to next-scale generation models like VAR (Tian et al., 2024) and Infinity (Han et al., 2025). More recently, models including Lumina-mGPT (Liu et al., 2024) and HunyuanImage (Cao et al., 2025) scale to tens of billions of parameters, substantially increasing inference cost (Feng et al., 2025). Mixture-of-Experts (MoE) architectures offer a scalable way to increase model capacity via sparse activation, and have been widely adopted in large language models (Dai et al., 2024; Xue et al., 2024; Lin et al., 2024; Yang et al., 2025). However, training MoE models from scratch remains expensive, which

---

[*]Equal contribution [1]Westlake University [2]Zhejiang University [3]Alibaba Group. Correspondence to: Huan Wang <wanghuan@westlake.edu.cn>.

*Proceedings of the 43rd International Conference on Machine Learning*, Seoul, South Korea. PMLR 306, 2026. Copyright 2026 by the author(s).

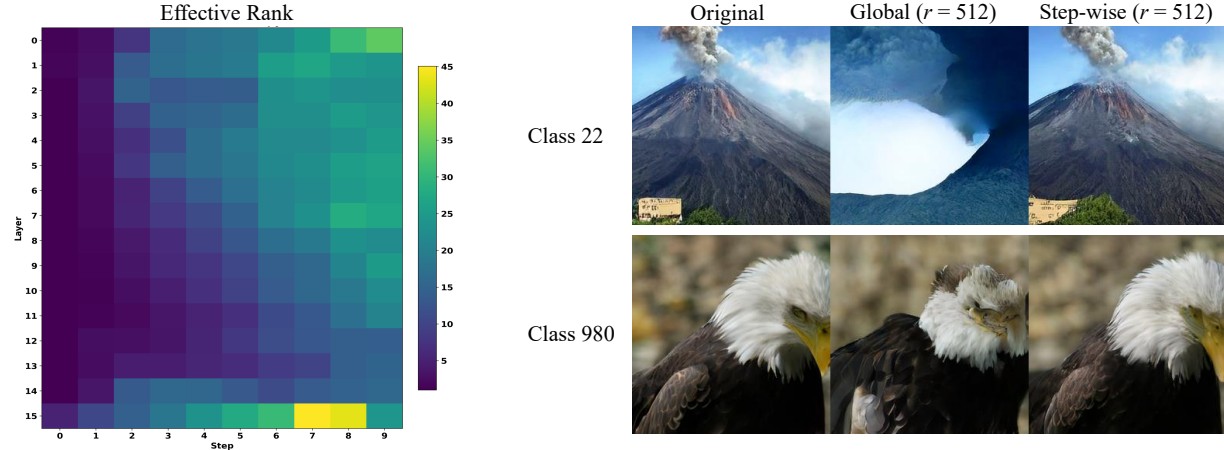

(a) Step-dependent effective rank of FFN activation across layers

(b) Visual comparison under global and step-wise low-rank constraints

*Figure 2.* **Motivation for trajectory-consistent decomposition.** PCA analysis of FFN activation patterns in a pretrained VAR-d16 model. (a) The effective rank of FFN activations varies drastically across generation scales, revealing that distinct feature subspaces dominate different stages of the generation trajectory. (b) Consequently, globally shared low-rank approximations can fail to model this evolving structure and may introduce severe artifacts, whereas step-wise approximations preserve generation fidelity. These observations indicate that explicitly preserving the *step-dependent activation trajectory* is crucial for high-quality expert initialization.

has motivated growing interest in low-cost conversion of pretrained dense models into sparse MoE variants (Komatsuzaki et al., 2023; Lee et al., 2024; Pei et al., 2025).

While active in LLMs, dense-to-MoE for autoregressive image generation is limited by dependencies and error accumulation. Vision methods (e.g., Dense2MoE (Zheng et al., 2025), MoLE (Zhu et al., 2024a), Diff-MoE (Cheng et al., 2025), CLIP-UP (Wang et al., 2025)) mainly target general upcycling or diffusion, often relying on heavy fine-tuning. Consequently, efficient conversion of pretrained autoregressive image generation models remains underexplored.

Motivated by these gaps, we study efficient dense-to-MoE conversion for autoregressive image generation and identify two key challenges: preserving step-dependent generation trajectories under strong activation non-stationarity during expert initialization, and enabling effective sparse fine-tuning beyond uniform load balancing given heterogeneous token uncertainty and spatial structure. To address these challenges, we propose **Prism-MoE**, a two-stage framework that combines trajectory-consistent expert initialization with confidence-adaptive sparse fine-tuning. With less than **10%** additional training budget, Prism-MoE enables sparse autoregressive models with only **37.5%** parameter activation to achieve generation quality comparable to dense counterparts across both class-to-image and text-to-image tasks.

**Contributions.** This work provides a systematic investigation of efficient dense-to-MoE conversion tailored to autoregressive image generation. Our contributions are threefold:

- We identify that standard dense-to-MoE initialization can fail to preserve step-dependent generation trajectories in autoregressive image models, and propose a

trajectory-consistent initialization that retains dense generation dynamics under sparse execution.

- We show that standard load-balancing objectives can be misaligned with autoregressive image generation, and introduce a confidence-adaptive sparse fine-tuning framework with a normalized cosine router and spatially structured regularization.

- Experiments across multiple backbones and tasks show that Prism-MoE provides strong dense-to-MoE initialization and, after efficient fine-tuning, enables models with only 37.5% parameter activation to achieve generation quality close to dense baselines using less than 10% additional training budget.

## 2. Related Work

### 2.1. Visual Autoregressive Modeling

Autoregressive image generation factorizes the joint distribution into a sequence of conditional predictions (Ramesh et al., 2021; Wang et al., 2024; Yu et al., 2025; Pang et al., 2025). Early pixel-level models (Oord et al., 2016; Van Den Oord et al., 2016) were limited by high dimensionality, while discrete visual tokens and Transformer backbones (Esser et al., 2021; Lee et al., 2022) enabled scalable token-based synthesis. Within this paradigm, raster-scan models such as LlamaGen (Sun et al., 2024) and Janus-Pro (Chen et al., 2025) perform next-token prediction on flattened 2D tokens, whereas VAR (Tian et al., 2024) and Infinity (Han et al., 2025) adopt next-scale generation (Zhang et al., 2024; Xie et al., 2024) from coarse to fine resolutions.

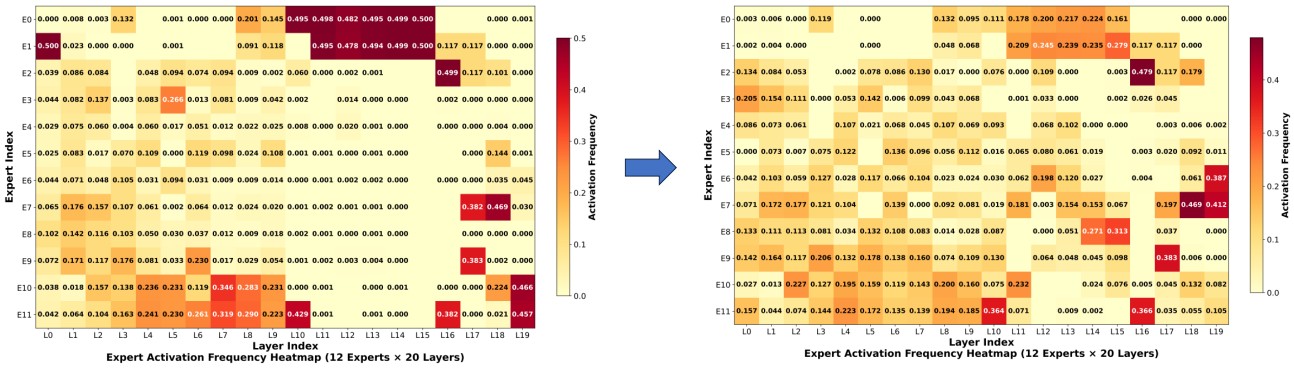

(a) After 5 epochs of fine-tuning (37.5% active, FID = 7.05)   (b) After 20 epochs of fine-tuning (37.5% active, FID = 10.7)

*Figure 3.* **Uniform load balancing can degrade expert specialization.** We initialize a dense-to-MoE variant of VAR-d20 with one shared and twelve routed experts (37.5% active parameters) and fine-tune it using a standard *load-balancing* loss. Despite increasingly uniform expert usage from epoch 5 to epoch 20, generation quality degrades (FID rises from 7.05 to 10.7), indicating that uniform routing supervision dilutes expert specialization and is misaligned with the intrinsic structure of visual tokens.

## 2.2. Dense-to-MoE Conversion

MoE enables conditional computation (Shazeer et al., 2017; Fedus et al., 2022), but training from scratch is costly (Komatsuzaki et al., 2023; Du et al., 2022; Dai et al., 2024), motivating dense-to-MoE upcycling. In language models, methods (e.g., CMoE (Pei et al., 2025), GMoE (Lee et al., 2024)) target expert initialization and routing stabilization. In vision, studies mainly target diffusion or general settings, such as Dense2MoE (Zheng et al., 2025), Jetpack (Zhu et al., 2024b), MoLE (Zhu et al., 2024a), Diff-MoE (Cheng et al., 2025), and CLIP-UP (Wang et al., 2025), relying on extensive distillation or fine-tuning to recover quality. Among these, Dense2MoE (Zheng et al., 2025) is most closely related, as it restructures diffusion Transformers into MoE models with shared and routed experts. However, diffusion generation differs substantially from visual autoregressive generation: AR generators expose step-wise non-stationarity, causal dependencies, and irreversible error accumulation, making trajectory preservation during sparse conversion more critical. Therefore, low-cost dense-to-MoE conversion for *autoregressive* visual generators remains largely unexplored, especially under sparse activation and limited fine-tuning budgets.

## 3. Method

This section presents the Prism-MoE framework, including empirical analysis of activation and routing dynamics (Sec. 3.1), trajectory-consistent initialization (Sec. 3.2), and confidence-adaptive sparse fine-tuning (Sec. 3.3).

### 3.1. Activation, Routing, and Token Uncertainty

To efficiently convert pretrained dense visual autoregressive models into Mixture-of-Experts architectures, two aspects are critical in practice: Effective expert initialization that fully leverages pretrained representations, and efficient fine-tuning that enables sparse activation without degrading gen-

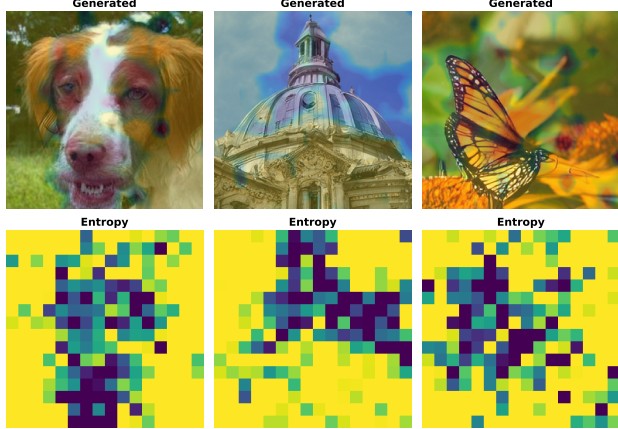

*Figure 4.* **Spatial entropy distribution of visual tokens. Top:** Dense model generations with entropy overlays, where high-entropy regions concentrate around semantically critical structures (e.g., object boundaries and fine details). **Bottom:** Token-level entropy heatmaps on the spatial token grid, revealing that uncertainty is highly non-uniform across tokens rather than evenly distributed.

eration quality. To study these two processes, we conduct two targeted analysis experiments.

**FFN activation structure.** We analyze FFN activations of pretrained dense VAR-d16 using PCA-based subspace characterization, comparing globally shared and step-wise low-rank approximations of the FC2 mapping (Fig. 2).

**Routing behavior under load balancing loss.** We examine router behavior in a dense-to-MoE model fine-tuned with standard load-balancing loss, analyzing expert activation frequencies and token-level routing statistics (Fig. 3).

**Token entropy distribution.** To characterize visual tokens, we analyze the spatial distribution of token-level entropy during dense autoregressive inference (Fig. 4).

**Observation 1.** *Trajectory consistency matters for expert initialization.* As shown in Fig. 2, PCA analysis reveals

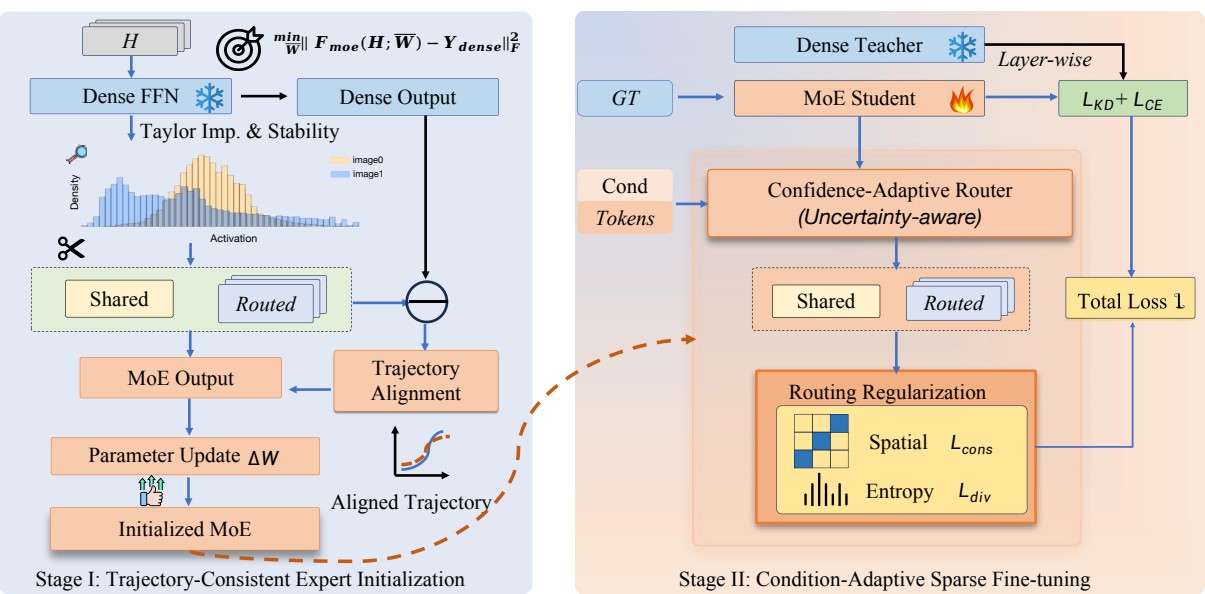

Figure 5. Overview of the Prism-MoE framework for efficient dense-to-MoE conversion of visual autoregressive models. **Left:** Stage I performs trajectory-consistent expert initialization by decomposing a pretrained dense FFN into shared and routed experts and aligning MoE outputs with the dense generation trajectory. **Right:** Stage II conducts condition-adaptive sparse fine-tuning, where a confidence-aware cosine router enables discriminative expert selection and spatially structured routing regularization promotes coherence and diversity under sparse activation. Together, these stages preserve generation fidelity while substantially reducing active parameters.

that FFN activation subspaces in pretrained VAR-d16 vary significantly across generation scales. Ignoring this non-stationarity, e.g., via a globally shared low-rank FC2 approximation, leads to severe quality degradation, whereas step-wise approximations aligned with the generation trajectory preserve fidelity close to the dense model. This indicates that effective initialization must respect the evolving activation structure along the autoregressive trajectory.

**Observation 2.** *Uniform load balancing suppresses expert specialization.* In a VAR-d20 dense-to-MoE model with 37.5% activation, standard load-balancing loss drives expert usage toward uniformity during fine-tuning, yet generation quality degrades (FID increases from 7.05 to 10.7). Token-level analysis (Appendix A) shows that most visual tokens have low entropy, causing routing probabilities to be weakly separated; as a result, the router satisfies the balancing objective via nearly uniform dispatch rather than token-dependent specialization.

**Observation 3.** *Visual entropy exhibits structural spatial inhomogeneity.* As shown in Fig. 4, token-level entropy in dense models is non-uniform across the image. High-entropy tokens concentrate around structurally or semantically salient regions (e.g., object boundaries), while most tokens lie in low-entropy areas. This spatial inhomogeneity calls for differentiated routing: high-entropy tokens require confident expert selection, whereas low-entropy regions can tolerate more balanced routing, motivating our confidence-adaptive and spatially-aware routing objective.

### 3.2. Trajectory-Consistent Initialization

We introduce a training-free, calibration-based initialization framework that converts a pretrained dense FFN into a sparse MoE FFN while preserving its generation behavior (Stage I in Fig. 5, left). The procedure uses hidden states collected from a small set of dense-model generation trajectories, but does not perform gradient-based training. Unlike prior dense-to-MoE approaches based on static neuron partitioning or clustering, our formulation enforces *trajectory consistency*, requiring the initialized MoE to reproduce the dense model's step-wise generation trajectory.

**Problem formulation.** Consider a pretrained dense FFN $\mathcal{F}_{\text{dense}}(\cdot; \mathbf{W})$ operating on a sequence of hidden states $\{\mathbf{h}^{(s)}\}_{s=1}^{S}$ collected along dense-model generation trajectories. Let $\mathbf{y}_{\text{dense}}^{(s)} = \mathcal{F}_{\text{dense}}(\mathbf{h}^{(s)}; \mathbf{W})$ denote the corresponding outputs. We initialize a sparse MoE FFN $\mathcal{F}_{\text{moe}}(\cdot; \widetilde{\mathbf{W}})$ under a predefined sparsity pattern, consisting of one shared expert and multiple routed experts, such that its outputs closely track the dense trajectory. Stacking all steps, we define $\mathbf{H} = [\mathbf{h}^{(1)}; \ldots; \mathbf{h}^{(S)}]$ and $\mathbf{Y}_{\text{dense}} = [\mathbf{y}_{\text{dense}}^{(1)}; \ldots; \mathbf{y}_{\text{dense}}^{(S)}]$, and formulate the objective as

$$\min_{\widetilde{\mathbf{W}}} \left\| \mathcal{F}_{\text{moe}}(\mathbf{H}; \widetilde{\mathbf{W}}) - \mathbf{Y}_{\text{dense}} \right\|_F^2, \qquad (1)$$

subject to the MoE sparsity constraints.

**Stage I: Structural decomposition.** We partition dense neurons based on a dual-metric criterion. Neuron importance is quantified via a first-order Taylor approximation:

$$\mathcal{I}_j = \mathbb{E}_{\mathbf{x}} \big| \mathbf{w}_j \odot \nabla_{\mathbf{w}_j} \mathcal{L}(\mathbf{x}) \big|. \tag{2}$$

This term estimates the contribution of each neuron to the dense model objective. To identify context-invariant features for the shared expert, we measure **activation stability** via the mean-to-variance ratio of post-activation responses across calibration trajectories. Neurons with high importance and stability are assigned to the shared expert, while remaining neurons are routed to task-specific experts.

**Stage II: Training-free trajectory alignment via constrained ridge compensation.** Structural decomposition alone inevitably introduces approximation errors. To improve trajectory fidelity without gradient-based fine-tuning, we perform a compensation step that explicitly aligns MoE outputs with the dense trajectory.

Let $\widetilde{\mathbf{W}}_0$ denote the parameters after Stage I, and $\mathbf{Y}_{\text{moe}}^{(0)} = \mathcal{F}_{\text{moe}}(\mathbf{H}; \widetilde{\mathbf{W}}_0)$ be the corresponding outputs. We define the residual matrix as

$$\mathbf{R} = \mathbf{Y}_{\text{dense}} - \mathbf{Y}_{\text{moe}}^{(0)}. \tag{3}$$

We freeze input projections and restrict compensation to output projections via a linear correction. Let $\mathbf{Z}$ denote the post-activation features feeding the output projection of a specific expert. We solve a constrained ridge regression problem:

$$\min_{\Delta\mathbf{W}} \|\mathbf{R} - \mathbf{Z}\Delta\mathbf{W}\|_F^2 + \lambda\|\Delta\mathbf{W}\|_F^2,$$
$$\text{s.t. } \|\Delta\mathbf{W}\|_F \leq \xi\|\mathbf{W}_{\text{old}}\|_F, \tag{4}$$

where $\lambda$ controls ridge regularization and $\xi$ defines a trust region relative to the original weight norm. The trust region prevents the closed-form compensation from excessively deviating from the original dense parameters. The unconstrained solution admits a closed form,

$$\Delta\mathbf{W}_{\text{ridge}} = (\mathbf{Z}^\top\mathbf{Z} + \lambda\mathbf{I})^{-1}\mathbf{Z}^\top\mathbf{R}, \tag{5}$$

which is projected back into the trust region via norm clipping. In practice, we adopt a *sequential* refinement strategy: the shared expert is first aligned to the initial residual $\mathbf{R}^{(0)}$, after which routed experts refine the remaining residual. This closed-form procedure improves step-wise trajectory alignment under sparse MoE execution, enabling high-fidelity expert initialization.

### 3.3. Condition-Adaptive Sparse Fine-tuning

After trajectory-consistent initialization, the fine-tuning stage comprises three components (Stage II in Fig. 5, right).

We apply layer-wise dense-to-sparse distillation to stabilize intermediate representations, adopt a magnitude-invariant routing network tailored for visual tokens, and introduce spatially structured routing regularization that accounts for visual correlation and token confidence. Together, these components preserve the dense teacher's generation behavior while encouraging spatially coherent and uncertainty-aware expert specialization beyond uniform load balancing.

**Dense-to-sparse distillation.** Fine-tuning is supervised by the standard cross-entropy loss and a layer-wise dense-to-sparse distillation objective. We perform distillation by matching hidden representations between the dense teacher and the MoE student using mean-squared error, applied every $k$ layers ($k=4$) with depth-increasing weights. The overall task loss is

$$\mathcal{L}_{\text{task}} = 0.5\,\mathcal{L}_{\text{CE}} + 0.5\,\mathcal{L}_{\text{KD}}. \tag{6}$$

This supervision preserves the dense teacher's intermediate trajectory while leaving routing specialization to the router and regularization terms.

**Normalized Cosine Router with Context Bias.** In visual autoregressive generation, token representations exhibit non-uniform magnitudes across spatial locations and generation stages, making magnitude-sensitive routing unstable. We therefore adopt a normalized cosine router that anchors expert selection on directional alignment.

Given a token representation $\mathbf{x}$ and a contextual condition $\mathbf{c}$, the routing logits are defined as

$$\mathbf{l}(\mathbf{x}, \mathbf{c}) = \left\langle \frac{\mathbf{x}}{\|\mathbf{x}\|_2}, \frac{\mathbf{W}}{\|\mathbf{W}\|_2} \right\rangle + g(\mathbf{x}, \mathbf{c}), \tag{7}$$

where the cosine similarity provides a magnitude-invariant semantic routing signal, and $g(\mathbf{x}, \mathbf{c})$ is a bounded auxiliary term that incorporates local token refinement and global contextual bias. This formulation reduces magnitude-induced routing fluctuation across spatial locations and generation stages, while retaining sufficient flexibility for contextual modulation. Routing probabilities are computed using sigmoid gating followed by top-$k$ selection,

$$p_e(\mathbf{x}) = \frac{\sigma(l_e)\,\mathbb{I}(e \in \text{top-}k)}{\sum_{j \in \text{top-}k} \sigma(l_j)}, \tag{8}$$

which avoids global softmax competition and enables independent expert evaluation.

**Structured regularization via uncertainty and spatial priors.** Standard MoE routing treats tokens independently and relies on global load balancing, ignoring the spatial structure of visual data. Visual tokens exhibit strong spatial correlation and heterogeneous uncertainty (Fig. 4), motivating structured priors on the routing distribution.

*Table 1.* **Main results on VAR-d20** (ImageNet-1K, class-conditional next-scale generation). We report efficiency (FLOPs) and image quality (FID, IS), evaluated with a batch size of 128. "w/ FT" indicates whether fine-tuning is applied, where N denotes no fine-tuning and Y denotes fine-tuning for 20 epochs.

| w/ FT | Method | Act. Params | FLOPs (TF) | FID↓ | IS↑ |
|---|---|---|---|---|---|
| N | Dense VAR-d20 | 100% | 240.6 | 2.96 | 304 |
| N | Magnitude Prune | 75% | 170.4 | 131 | 8.22 |
| N | Importance Prune (Taylor) | 75% | 170.4 | 75.3 | 27.0 |
| N | Jetpack | 75% | 170.4 | 131 | 8.22 |
| N | CMoE | 75% | 170.4 | 22.8 | 121 |
| N | GMoE | 75% | 170.4 | 63.7 | 67.6 |
| N | Dense2MoE | 75% | 170.4 | 9.82 | 216 |
| N | **Prism-MoE (ours)** | 75% | 170.4 | **5.03** | **255** |
| Y | Jetpack | 37.5% | 136.2 | 163 | 5.22 |
| Y | Dense2MoE | 37.5% | 136.2 | 9.03 | 248 |
| Y | Linear Softmax Router + LB Loss | 37.5% | 136.2 | 10.7 | 247 |
| Y | **Prism-MoE (ours)** | 37.5% | 136.2 | **2.86** | **303** |

*Spatial smoothness with uncertainty-adaptive regularization.* We treat routing probabilities as a spatially structured field and encourage local consistency more strongly for uncertain tokens. Let $\mathbf{p}_i \in \Delta^E$ denote the routing distribution of token $i$, and let $\mathcal{N}(i)$ denote its spatial neighborhood. We define a locally aggregated reference distribution

$$\mathbf{q}_i = \frac{1}{|\mathcal{N}(i)|} \sum_{j \in \mathcal{N}(i)} \mathbf{p}_j. \tag{9}$$

The spatial consistency loss is formulated as a confidence-weighted KL divergence:

$$\mathcal{L}_{\text{cons}} = \mathbb{E}_i \big[ (1 - w_i) \cdot \text{KL}(\mathbf{p}_i \,\|\, \mathbf{q}_i) \big], \tag{10}$$

where $w_i \in [0, 1]$ is a normalized confidence score derived from the dense teacher's logit margin. Low-confidence tokens receive stronger local consistency regularization to reduce unstable routing, while high-confidence tokens are allowed to form sharper, semantically meaningful routing transitions.

*Spatial neighborhood definition.* $\mathcal{N}(i)$ is defined on the 2D token grid. For flattened autoregressive sequences, the spatial layout is reconstructed during training solely for regularization, without affecting causal inference.

*Global diversity via marginal entropy.* To avoid expert under-utilization without suppressing confident token-level routing, we regularize the batch-level marginal expert distribution over all tokens in a batch $\mathcal{B}$:

$$\bar{u}_e = \frac{1}{|\mathcal{B}|} \sum_{i \in \mathcal{B}} p_{e,i}, \tag{11}$$

by minimizing its negative entropy:

$$\mathcal{L}_{\text{div}} = 1 - \frac{-\sum_e \bar{u}_e \log \bar{u}_e}{\log E}. \tag{12}$$

This marginal regularization promotes batch-level expert utilization, rather than forcing each token to use a uniform expert distribution.

The final objective is

$$\mathcal{L} = \mathcal{L}_{\text{task}} + \lambda_1 \mathcal{L}_{\text{cons}} + \lambda_2 \mathcal{L}_{\text{div}}. \tag{13}$$

It combines task supervision with spatial and diversity priors, improving routing stability and expert utilization without directly overriding the dense teacher's semantic guidance.

## 4. Experiments

### 4.1. Experimental Setup

**Training setup.** All models are fine-tuned on NVIDIA A100 GPUs. For fair comparison, all MoE variants are trained under the same hardware and optimization settings.

**Backbones.** We evaluate Prism-MoE on two visual autoregressive backbones: VAR-d20 (Tian et al., 2024), which follows a *next-scale* paradigm for class-to-image (C2I) generation, and LlamaGen-XL (Sun et al., 2024), a *next-token* model for text-to-image (T2I) generation. LlamaGen-XL-T2I is trained in two stages on LAION-COCO (Schuhmann et al., 2022) and an internal dataset; we use the Stage I model for fine-tuning and quantitative evaluation, while Stage II is only used for qualitative visualization.

**Datasets.** VAR-d20 is both trained and evaluated on ImageNet-1K (Deng et al., 2009), while LlamaGen-XL is fine-tuned on 100K samples from LAION-COCO (Schuhmann et al., 2022) and evaluated on the MS-COCO 2017 validation set (Lin et al., 2014).

**Baselines and comparison methods.** We evaluate expert initialization and fine-tuning separately.

*Expert initialization.* All MoE variants are compared under

*Table 2.* **Main results on LlamaGen-XL-T2I-Stage I**. We report efficiency (GFLOPs) and image quality (FID, CLIP-Score).

| w/ FT | Method | Act. Params | GFLOPs | FID↓ | CLIP-Score↑ |
|---|---|---|---|---|---|
| N | Dense LlamaGen-XL (Stage I) | 100% | 1175 | 25.8 | 32.0 |
| N | Magnitude Prune | 75% | 991 | 54.5 | 25.0 |
| N | Importance Prune (Taylor) | 75% | 991 | 27.4 | 30.8 |
| N | Dense2MoE | 75% | 991 | 42.8 | 17.6 |
| N | **Prism-MoE (ours)** | 75% | 991 | **27.1** | **31.2** |
| Y | Linear Softmax Router + LB Loss | 37.5% | 712 | 31.5 | 27.9 |
| Y | **Prism-MoE (ours)** | 37.5% | 712 | **26.0** | **32.1** |

*Table 3.* **End-to-end inference latency.** Wall-clock latency is measured on a single A100 with batch size 128. Prism-MoE uses a custom Triton fused MoE backend following vLLM-style grouped expert execution (Kwon et al., 2023; Gale et al., 2023).

| Backbone | Model | Latency ↓ | Speedup ↑ |
|---|---|---|---|
| VAR-d20 | Dense | 45.5 ms | 1.00× |
| | Prism-MoE | 37.2 ms | 1.22× |
| LlamaGen-XL | Dense | 293.6 ms | 1.00× |
| | Prism-MoE | 237.6 ms | 1.24× |

*Table 4.* **Component-wise ablation study on VAR-d16.** We evaluate the contribution of each component in Prism-MoE. The baseline (Row 1) uses standard importance pruning without trajectory alignment. Rows 3-5 include fine-tuning for 20 epochs. **Prism-MoE** (Row 5) integrates all proposed methods.

| w/ FT | Configuration | Act. | FID↓ |
|---|---|---|---|
| *Phase 1: Initialization Effectiveness* | | | |
| N | Init (Taylor Importance) | 75% | 27.5 |
| N | **+ Trajectory-Consistent Init** | 75% | 12.6 |
| *Phase 2: Fine-tuning Components* | | | |
| Y | + Fine-tuning (Distillation only) | 37.5% | 9.00 |
| Y | + Normalized Cosine Router | 37.5% | 6.04 |
| Y | + Spatial Loss (**Full Prism-MoE**) | 37.5% | **4.26** |

a fixed 75% parameter activation ratio. We include representative dense-to-MoE methods, including Jetpack (Zhu et al., 2024b), CMoE (Pei et al., 2025), GMoE (Lee et al., 2024), and Dense2MoE (Zheng et al., 2025).

*Static Pruning.* We compare with two methods, Magnitude (Han et al., 2015) and First-Order Taylor (Molchanov et al., 2019), which select neurons based on weight magnitude and weight–gradient products, respectively.

*Fine-tuning.* For routing and optimization comparison, all methods are fine-tuned from the same VAR-d20 MoE initialization for 20 epochs, corresponding to approximately 8% of the training cost of dense training from scratch.

Evaluation metrics. We assess both generation quality and efficiency. Image quality is measured using FID and Incep-

tion Score (IS), while efficiency is evaluated in terms of FLOPs and end-to-end latency under sparse execution.

**Implementation Details.** More details are provided in the Appendix B. Code is available at `https://github.com/NeuraLiying/Prism-MoE`.

### 4.2. Main Results

**Class-to-Image (C2I).** For the class-to-image (C2I) task, Table 1 reports results on VAR-d20 under both initialization-only and fine-tuning settings. Without fine-tuning (w/ FT = N), all sparse methods are evaluated at the same 75% activation ratio. Compared to pruning-based baselines and existing MoE initializations, Prism-MoE yields the lowest FID and the highest IS among all sparse variants, which indicates a higher-quality dense-to-MoE initialization.

We further evaluate fine-tuned models under a more aggressive activation ratio of 37.5% (w/ FT = Y). After 20 epochs of fine-tuning, Prism-MoE outperforms the evaluated competing methods, achieving both lower FID and higher IS, while using substantially fewer activated parameters and FLOPs. These results demonstrate that the proposed initialization not only provides strong performance without fine-tuning, but also remains effective after fine-tuning, enabling high-quality image generation under limited budgets.

**Text-to-Image (T2I).** We evaluate Prism-MoE on the LlamaGen-XL-T2I-Stage I backbone, with results summarized in Table 2. Without fine-tuning (w/ FT = N), Prism-MoE achieves better generation quality than pruning-based methods and Dense2MoE under the same 75% activation ratio, yielding lower FID and higher CLIP-Score, and closely matching the dense baseline. After fine-tuning with a more aggressive activation ratio of 37.5% (w/ FT = Y), Prism-MoE further improves generation quality, achieving the best FID and CLIP-Score among all sparse variants, demonstrating its effectiveness for text-to-image generation under tight compute budgets.

**End-to-end latency.** We further evaluate wall-clock speedup using our Prism-MoE router with a custom Tri-

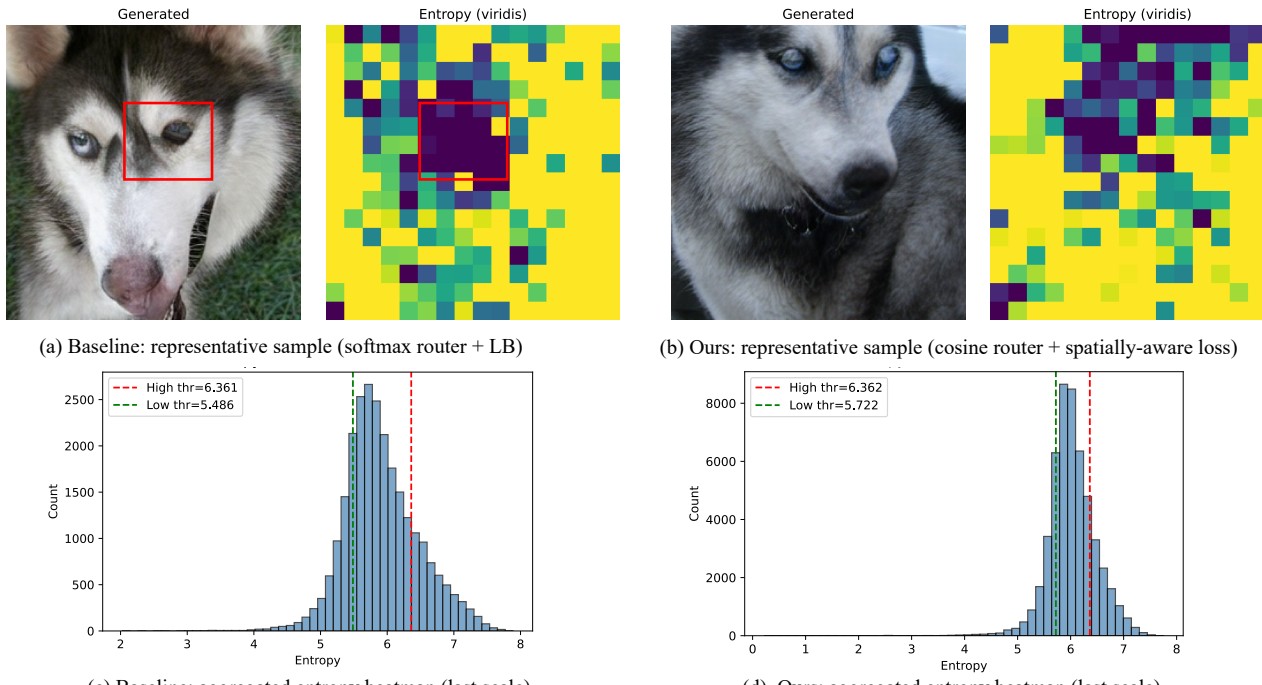

(a) Baseline: representative sample (softmax router + LB)    (b) Ours: representative sample (cosine router + spatially-aware loss)

(c) Baseline: aggregated entropy heatmap (last scale)    (d) Ours: aggregated entropy heatmap (last scale)

*Figure 6. Cosine-similarity router and spatially-aware loss yield improved qualitative samples and a reduced high-entropy tail.* (a) Baseline sample generated with a softmax router and load-balancing loss, showing structural distortion (highlighted with a red box). (b) Sample generated by the proposed fine-tuning under the same class label and random seed. (c,d) Token-level entropy heatmaps at the last scale for the baseline and the proposed method. Bottom: entropy histograms at the last scale for the two settings.

ton fused MoE backend following vLLM-style grouped expert execution (Kwon et al., 2023; Gale et al., 2023). As shown in Table 3, Prism-MoE achieves 1.22× speedup on VAR-d20 and 1.24× speedup on LlamaGen-XL. All measurements use a single A100 80G with batch size 128; distributed all-to-all communication is not included.

### 4.3. Ablation Study

We conduct a component-wise ablation study on the VAR-d16 backbone to analyze the contribution of each design choice in Prism-MoE, as shown in Table 4. Starting from a baseline initialized with Taylor-based importance pruning, introducing the proposed trajectory-consistent initialization leads to a substantial reduction in FID, indicating a significantly improved expert initialization. We then fine-tune the model under a lower activation ratio of 37.5%. Fine-tuning with distillation alone further improves generation quality, while incorporating the normalized cosine router yields additional gains. Finally, integrating the spatially-aware loss results in the best performance, corresponding to the full Prism-MoE configuration. These results demonstrate that each component consistently contributes to improved generation quality, and their combination is crucial for achieving optimal performance under sparse activation.

**Entropy-based qualitative ablation.** As a qualitative complement to Table 4, we compare the baseline fine-tuning

setup (softmax router with load-balancing loss) with the full fine-tuning objective that includes the cosine-similarity router and the spatially-aware loss, using identical generation conditions. Figure 6 shows that the baseline exhibits a noticeable structural distortion (red box) in regions where high-entropy tokens form dense spatial clusters, which correspond to more challenging and informative generation decisions. With the cosine router and spatially-aware loss enabled, the generated sample is visually improved and the high-entropy patterns become more localized. This behavior is consistent with the entropy histograms, where the proposed fine-tuning reduces the high-entropy tail. Additional examples are provided in the appendix.

**Broader dense-to-MoE initialization.** We further examine whether the proposed trajectory-consistent initialization transfers beyond the two main backbones. Semanticist (Wen et al., 2025) evaluates transfer to a hybrid AR-diffusion architecture, while Infinity-8B (Han et al., 2025) evaluates scalability to a larger 8B backbone. Table 5 reports initialization-only results without fine-tuning. Prism-MoE remains close to the dense baseline in both settings, providing preliminary evidence for broader applicability.

**Sensitivity analysis.** We study the robustness of Prism-MoE to calibration and ridge regularization choices. As shown in Table 6, Prism-MoE yields stable performance across different calibration set sizes and ridge multipliers

*Table 5.* **Broader dense-to-MoE initialization evaluation.** We evaluate Prism-MoE initialization on Semanticist and Infinity-8B without fine-tuning.

| Backbone | Metric | Act. | Dense | Prism-MoE |
|---|---|---|---|---|
| Semanticist | FID ↓ | 75% | 2.49 | 2.99 |
| Infinity-8B | GenEval ↑ | 75% | 0.80 | 0.78 |
| Infinity-8B | DPG-Bench ↑ | 75% | 86.6 | 86.4 |

*Table 6.* **Sensitivity analysis.** We evaluate Prism-MoE under different calibration sizes and ridge multipliers.

| Factor | Setting | VAR-d20 FID ↓ | LlamaGen FID ↓ |
|---|---|---|---|
| Calib. size | 100 | 6.04 | 27.4 |
| | 200 | 5.03 | 27.1 |
| | 300 | 4.96 | 27.0 |
| Ridge mult. | 0.5 | 5.06 | 27.0 |
| | 1.0 | 5.03 | 27.1 |
| | 2.0 | 5.28 | 27.3 |

on both VAR-d20 and LlamaGen. These results suggest that trajectory-consistent initialization is reasonably robust under the tested settings.

### 4.4. Generation Quality

We present qualitative comparisons to visually assess the generation quality under different sparsity and fine-tuning settings. For VAR-d20, we show results from the original dense model, the 75% activated Prism-MoE without fine-tuning, and the 37.5% activated Prism-MoE with fine-tuning. For LlamaGen-XL, we include visual results from the Stage I backbone under the original dense setting, the 75% activated model without fine-tuning, and the 37.5% activated model with fine-tuning. In addition, we provide visualizations from the LlamaGen-XL Stage II dense model with 75% activation and without fine-tuning to further illustrate generation behavior at a later training stage.

Fig. 7 presents qualitative visualizations of VAR-d20 with 37.5% parameter activation after fine-tuning. The figure provides representative generated samples under the low-activation setting, illustrating the generation behavior of Prism-MoE in this regime. We further include qualitative samples from LlamaGen-XL Stage I and Stage II with 75% activation after Prism-MoE initialization to illustrate generation behavior across training stages (Fig. 8).

## 5. Conclusion

In this work, we study efficient dense-to-MoE conversion for visual autoregressive generation, where step-wise dependency, non-stationary generation trajectories, and irreversible error accumulation make sparse modeling particularly challenging. We identify key limitations in existing dense-to-MoE strategies, including trajectory mismatch dur-

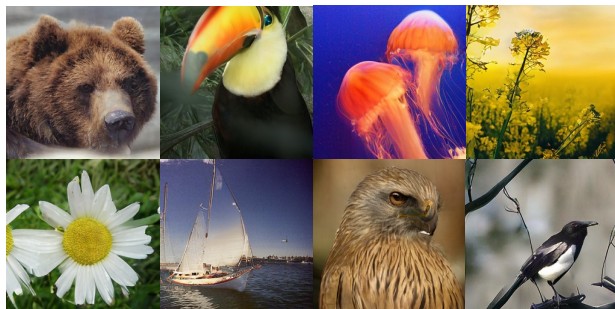

*Figure 7.* Qualitative visualization of VAR-d20 with 37.5% parameter activation after fine-tuning. The figure shows representative generated samples under identical generation settings, illustrating the visual characteristics of Prism-MoE at a low activation ratio.

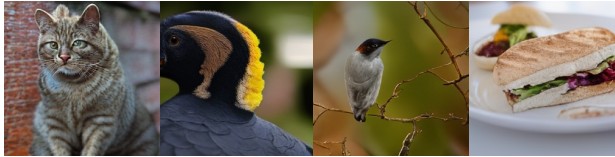

(a) LlamaGen-XL Stage I (75% activation w/o FT)

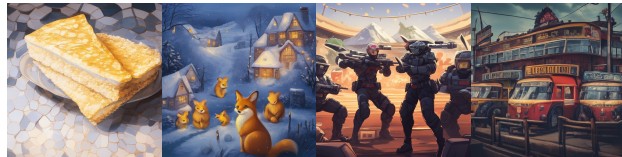

(b) LlamaGen-XL Stage II (75% activation w/o FT)

*Figure 8.* Qualitative results of LlamaGen-XL with 75% parameter activation after Prism-MoE initialization. (a) Stage I. (b) Stage II.

ing initialization and misaligned routing objectives during fine-tuning. To address these issues, we propose Prism-MoE, a two-stage framework that combines trajectory-consistent expert initialization with condition-adaptive sparse fine-tuning. Experiments on next-scale and next-token visual autoregressive generators show that Prism-MoE achieves near-dense generation quality with substantially reduced active parameters and limited additional training cost. Additional latency, sensitivity, and broader initialization evaluations further support its practical efficiency and robustness under the tested settings.

## Acknowledgements

This paper is supported by Young Scientists Fund of the National Natural Science Foundation of China (NSFC) (No. 62506305), Zhejiang Leading Innovative and Entrepreneur Team Introduction Program (No. 2024R01007), Key Research and Development Program of Zhejiang Province (No. 2025C01026), Scientific Research Project of Westlake University (No. WU2025WF003), Chinese Association for Artificial Intelligence (CAAI) & Ant Group Research Fund - AGI Track (No. 2025CAAI-ANT-13). It is also supported by the research funds of the National Talent Program and Hangzhou Municipal Talent Program.

## Impact Statement

This paper improves the efficiency of visual autoregressive generation through dense-to-MoE conversion. The method is mainly evaluated on visual autoregressive backbones, with preliminary evidence on broader and larger models; broader generalization and more aggressive sparsity remain to be further studied. Practical speedups depend on fused MoE kernels and hardware-aware implementation. More efficient image generation may reduce the cost of synthetic media production, which can support benign applications but may also increase misuse risks such as misinformation and deepfakes. Responsible deployment should follow existing safeguards for generative models.

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

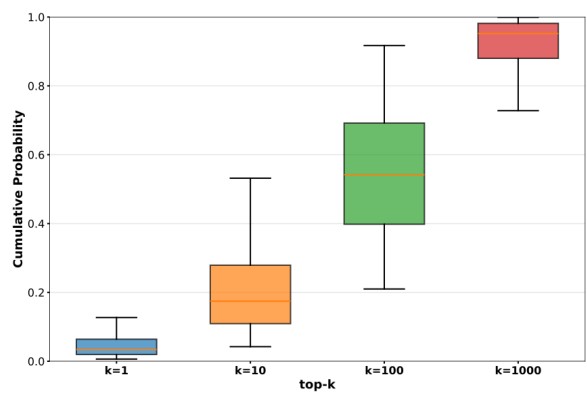 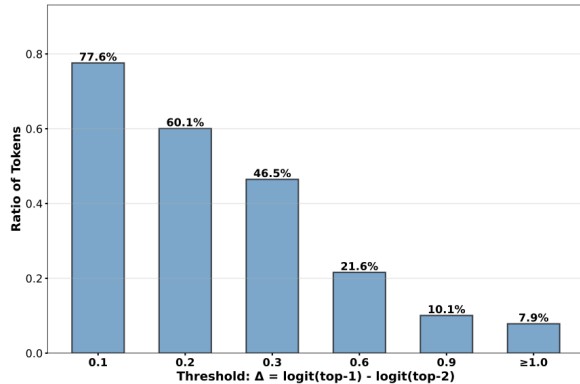

(a) Cumulative probability mass under top-k logits   (b) Distribution of logit margin between top-1 and top-2 predictions

*Figure 9.* **Token-level logit characteristics of dense visual autoregressive models.** Vocabulary-level logit statistics collected during inference of a pretrained VAR-d16 model. (a) Cumulative probability mass captured by the top-$k$ logits after softmax, showing that for most visual tokens the probability mass is weakly concentrated. (b) Distribution of logit margins, defined as the difference between the highest and second-highest logits, indicating limited separation between competing predictions for most tokens.

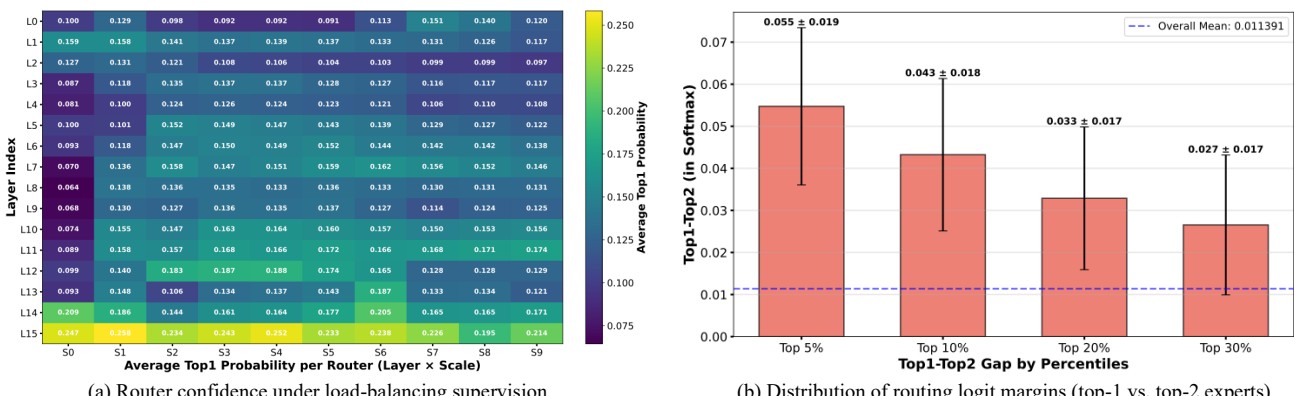

(a) Router confidence under load-balancing supervision   (b) Distribution of routing logit margins (top-1 vs. top-2 experts)

*Figure 10.* **Router behavior under load-balancing supervision.** (a) Average top-1 routing probability across layers and generation scales, showing that routing confidence remains broadly uniform without clear stage-wise preference. (b) Distribution of the routing margin between the top-1 and top-2 experts, indicating generally small separations and thus weak token-level routing decisiveness.

## A. Limitations of Load-Balancing Loss in Autoregressive Visual MoE

**Routing uncertainty under load-balancing supervision.**   We first examine the intrinsic uncertainty of visual tokens in a pretrained dense VAR-d16 model by analyzing its output logits during inference. As shown in Fig. 9, a large fraction of tokens exhibit weak probability concentration, reflected by low top-$k$ cumulative mass and small logit margins. This indicates that many visual tokens are inherently ambiguous, with multiple candidates remaining similarly plausible.

Such low-confidence token representations provide *limited discriminative signal for routing*. Under this condition, routing objectives that emphasize uniform expert usage can easily dominate optimization. As shown in Fig. 10, a router trained with a standard load-balancing loss exhibits consistently low routing decisiveness. The margin between the top-1 and top-2 experts remains small for most tokens, and this behavior persists across layers and generation scales, indicating that the router satisfies the balancing constraint through near-uniform dispatch rather than meaningful token-dependent selection.

These results explain why load-balancing supervision can prevent expert collapse yet still degrade generation quality: expert usage becomes uniform, but specialization is suppressed due to weak and uninformative routing decisions.

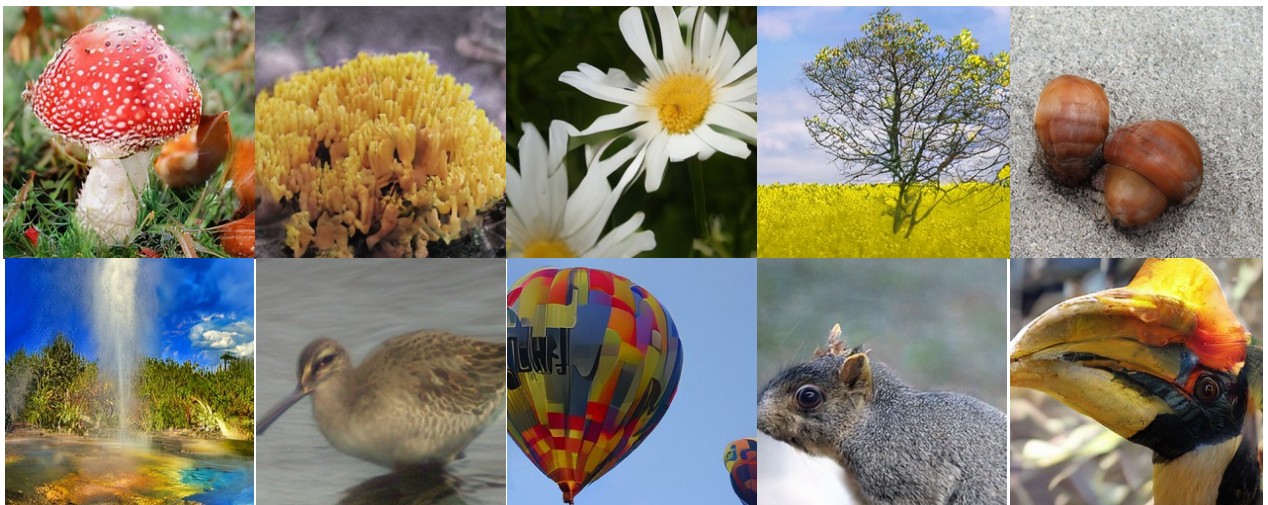

*Figure 11.* Additional visual results on VAR-d20 with 75% active parameters, evaluated without fine-tuning (w/o FT).

## B. Experimental Details

**Expert partitioning.**    For dense-to-MoE conversion, we partition each dense FFN into one shared expert and twelve routed experts. The shared expert is assigned 25% of the original FFN hidden dimension, while the remaining capacity is evenly distributed across the routed experts. This design follows the shared–routed expert paradigm adopted in DeepSeek-MoE (Dai et al., 2024) and is further informed by the expert partitioning analysis in Dense2MoE (Zheng et al., 2025). We adopt this configuration as a balanced choice that supports stable shared feature extraction while enabling effective expert specialization under sparse routing.

**Evaluation settings.**    For VAR-d20, image generation quality is evaluated under the class-to-image (C2I) setting using classifier-free guidance with CFG $= 1.5$ and temperature $= 1.0$. For LlamaGen-XL, both Stage I and Stage II models are evaluated under the text-to-image (T2I) setting with CFG $= 7.5$, top-$k = 2000$, top-$p = 1.0$, and temperature $= 1.0$.

For ImageNet C2I evaluation, we generate 50,000 samples following standard practice. For LlamaGen-XL, 5,000 images are generated from prompts in the MS-COCO 2017 validation set for quantitative evaluation.

**Initialization calibration.**    For Stage I initialization, we use a calibration set of 200 samples for VAR-d20, LlamaGen-XL Stage I, and LlamaGen-XL Stage II. The calibration samples are randomly drawn from the training set with diverse class labels or text prompts. All dense-to-MoE initialization methods use the same calibration protocol to ensure fair comparison.

## C. More Visual Results

This section provides additional qualitative comparisons for Prism-MoE on VAR-d20 and LlamaGen-XL. All results are generated under the same sampling configurations as described in Appendix B.

### C.1. VAR-d20, 75% Active Parameters, without Fine-tuning

Fig. 11 shows qualitative results on VAR-d20 under a 75% parameter activation ratio without fine-tuning (w/o FT), highlighting the effect of Stage I initialization.

### C.2. VAR-d20, 37.5% Active Parameters, with Fine-tuning

Fig. 12 shows results on VAR-d20 under a 37.5% parameter activation ratio with fine-tuning (w/ FT), demonstrating that sparse fine-tuning further improves visual fidelity.

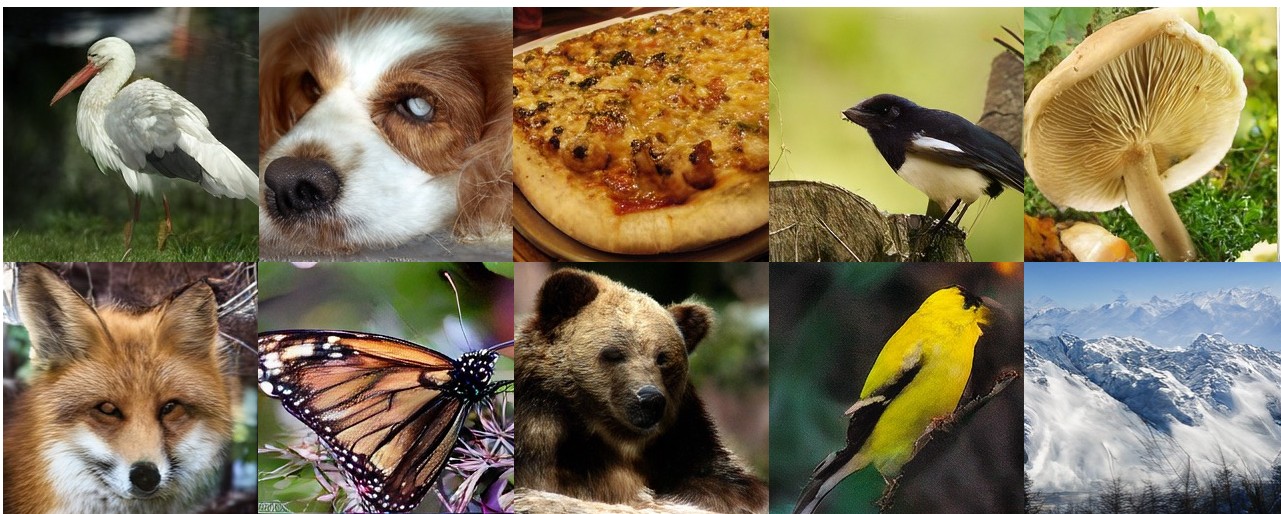

*Figure 12.* Additional visual results on VAR-d20 with 37.5% active parameters, evaluated with fine-tuning (w/ FT).

### C.3. LlamaGen-XL Stage I, 37.5% Active Parameters, with Fine-tuning

Fig. 13 reports qualitative text-to-image generations from LlamaGen-XL Stage I under a 37.5% parameter activation ratio with fine-tuning (w/ FT).

## D. Discussion and Future Work

Prism-MoE demonstrates that dense-to-MoE conversion can be an effective and low-cost strategy for visual autoregressive models. By preserving generation trajectories during initialization and enabling stable sparse fine-tuning, the proposed framework achieves high-quality generation under aggressive sparsity without retraining from scratch. These results indicate that trajectory-aware conversion and uncertainty-guided routing are key ingredients for practical MoE deployment in visual generation. Looking forward, the ideas in Prism-MoE may be extended beyond FFN modules to attention or hybrid components in autoregressive models. Applying the framework to larger-scale or multimodal generative models is another promising direction for future exploration.

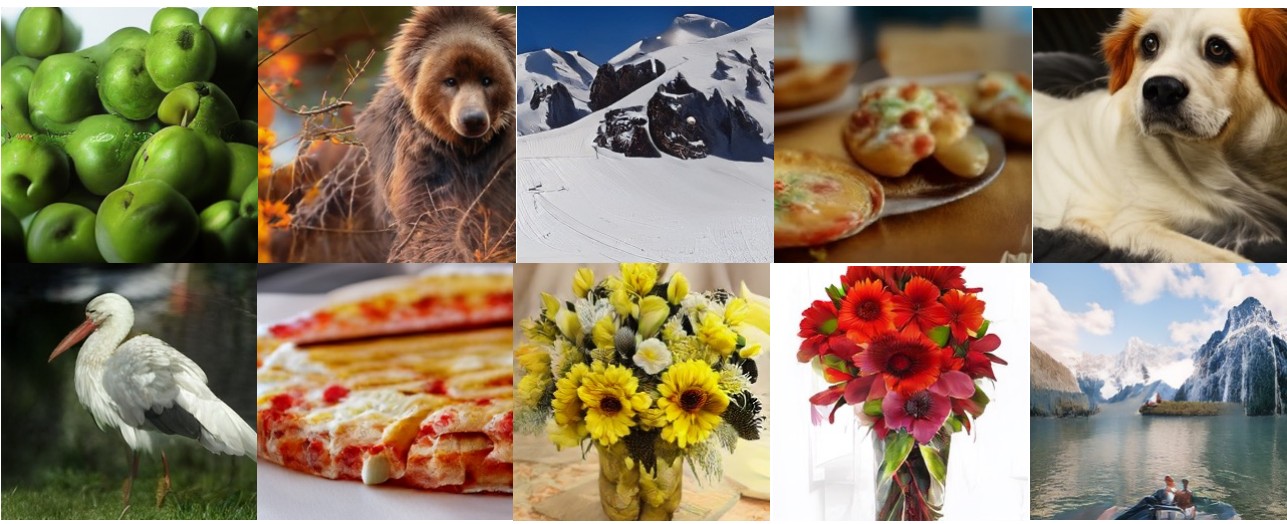

*Figure 13.* Additional visual results on LlamaGen-XL Stage I with 37.5% active parameters, evaluated with fine-tuning (w/ FT).

