# OpenReview forum: "Prism-MoE: Efficient Dense-to-MoE Conversion for Visual Autoregressive Generation"
_ICML.cc/2026/Conference — ICML 2026 regular_

### Official Review · Reviewer_Y9cC · 2026-03-10

**Soundness:** 3
**Presentation:** 3
**Significance:** 2
**Originality:** 3
**Overall Recommendation:** 4
**Confidence:** 3

**Summary:**

This paper proposes Prism-MoE, a framework for converting pretrained dense visual autoregressive models into sparse Mixture-of-Experts (MoE) architectures with minimal additional training. While dense autoregressive models achieve strong image generation performance, they require substantial computational resources, and training MoE models from scratch is also costly. Prism-MoE addresses this by introducing a dense-to-MoE upcycling strategy that preserves the behavior of the original dense model.

The method includes two stages. First, a trajectory-consistent initialization partitions dense feed-forward neurons into shared and routed experts based on importance and activation stability, followed by a ridge-regression alignment step to minimize the discrepancy between dense and MoE outputs. Second, a confidence-adaptive sparse fine-tuning stage improves expert routing using cosine-based routing, spatial routing regularization for visual tokens, and an expert diversity constraint to avoid expert collapse.

Experiments on VAR and LlamaGen demonstrate that Prism-MoE maintains generation quality while significantly reducing the number of active parameters during inference.

**Compliance With Llm Reviewing Policy:**

Affirmed.

**Final Justification:**

Although the algorithmic novelty is somewhat incremental, the authors’ comprehensive rebuttal has effectively addressed my empirical concerns regarding scalability and latency, which leads me to raise my score, as the proposed dense-to-MoE conversion framework is technically sound, interesting, and practically useful.

**Key Questions For Authors:**

1.Scalability to Larger Models.
The experiments are conducted on models such as VAR-d20 and LlamaGen-XL, which are relatively moderate in scale compared with the very large generative models where MoE architectures are typically most beneficial. Could the authors provide evidence that the proposed dense-to-MoE conversion framework scales effectively to significantly larger models (e.g., multi-billion parameter architectures)? In particular, it would be helpful to understand whether the trajectory-consistent initialization and routing strategies remain stable when applied to larger networks. A convincing demonstration of scalability would strengthen the practical significance of the work and would positively influence my overall evaluation of the paper.

2.Practical Efficiency and Inference Latency.
The paper mainly reports improvements in terms of active parameters and generation quality, but does not provide detailed measurements of real-world inference efficiency, such as latency, throughput, or communication overhead. Since MoE architectures often introduce routing and communication costs that can affect deployment efficiency, could the authors report system-level performance metrics under realistic inference settings? Additional evidence showing that the proposed approach provides tangible speed or efficiency gains would increase confidence in its practical value.

3.Sensitivity to Hyperparameters and Initialization Choices.
The trajectory-consistent initialization relies on several design choices, such as the criteria for neuron partitioning (importance and activation stability) and the ridge regression alignment step. How sensitive is the final performance to these choices and to the associated hyperparameters? For example, if the partition ratio between shared and routed experts changes, or if the regularization strength in the alignment step varies, does the model remain stable? A clearer understanding of the robustness of the initialization procedure would help assess the reliability and reproducibility of the method.

4.Generalization Beyond Visual Autoregressive Models.
The proposed framework is specifically designed for visual autoregressive generation models. To what extent can the method generalize to other types of architectures, such as diffusion models, multimodal transformers, or large language models? Even preliminary evidence or discussion regarding broader applicability would help clarify whether the contribution represents a general dense-to-MoE conversion strategy or a specialized solution for a specific model family. If the approach can be generalized more broadly, this would increase the perceived impact of the work.

**Limitations:**

No.

The paper only provides a very brief statement in the broader impact section and does not meaningfully discuss the limitations of the proposed approach or the potential societal implications of improving the efficiency of large generative models. A more thorough discussion would strengthen the transparency and responsibility of the work.

First, the authors should more explicitly describe the technical limitations of their method. For example, the proposed dense-to-MoE conversion is evaluated only on a limited set of visual autoregressive models and moderate model scales. It remains unclear how well the approach generalizes to substantially larger architectures or to other generative paradigms. The authors could also discuss potential sensitivity to hyperparameters, expert allocation strategies, or routing configurations, as well as situations where the trajectory-consistent initialization might fail to preserve generation quality.

Second, the paper could benefit from a clearer discussion of practical deployment limitations. Although MoE architectures reduce the number of active parameters, they may introduce additional routing overhead and communication costs in distributed environments. Discussing these system-level considerations would provide a more balanced view of the method’s practical advantages and constraints.

Finally, the authors should briefly consider potential societal risks associated with improving the efficiency of large-scale image generation models. More efficient generative systems could lower the barrier to producing large volumes of synthetic media, which may increase the risk of misuse such as misinformation, deepfakes, or large-scale automated content generation. A short discussion acknowledging these risks and emphasizing responsible use would improve the completeness of the broader impact section.

**Strengths And Weaknesses:**

**Strengths**
The paper addresses a relevant problem of converting dense visual autoregressive models into sparse Mixture-of-Experts (MoE) architectures to reduce computational cost. The proposed two-stage framework—trajectory-consistent initialization followed by confidence-adaptive sparse fine-tuning—provides a reasonable solution for maintaining generation stability during dense-to-MoE conversion. The initialization strategy, which partitions feed-forward neurons based on importance and activation stability, is logically motivated, and the ridge-regression alignment step offers a practical way to reduce discrepancies between dense and MoE outputs.

Empirically, experiments on multiple autoregressive image generation models demonstrate that the proposed method can preserve generation quality while activating fewer parameters during inference. The evaluation includes several relevant baselines and ablation studies, which generally support the effectiveness of the approach.

The paper is also clearly written and well structured. The motivation, methodological design, and experimental validation are presented in a coherent manner, and the figures help illustrate the framework and routing mechanisms. Overall, the work addresses a timely problem in improving the efficiency of large-scale visual generative models and provides a practical dense-to-MoE conversion strategy.


**Weaknesses**
Although the proposed framework is technically reasonable, the level of methodological novelty appears somewhat limited. Several components, such as neuron importance-based partitioning, ridge-regression alignment, and cosine-based routing, resemble existing techniques in pruning, output matching, and MoE routing design. As a result, the contribution mainly lies in integrating and adapting known methods rather than introducing fundamentally new algorithmic ideas.

The empirical evaluation, while generally adequate, could be further strengthened. Experiments are conducted on a limited set of autoregressive generation models, and scalability to significantly larger architectures is not thoroughly examined. Since MoE methods are particularly valuable at very large model scales, additional large-scale evaluations would better demonstrate the practical impact of the approach. Moreover, the paper mainly reports generation quality and parameter efficiency, while system-level aspects such as inference latency and deployment efficiency are not analyzed.

Finally, although the paper is clearly written, the discussion of theoretical motivation, limitations, and potential failure cases could be expanded. The generalizability of the framework beyond visual autoregressive models also remains unclear.

---

> ### Author Rebuttal · Authors · 2026-03-31
>
> We thank the reviewer for the valuable comments, and we respond to each point below.
>
>
> **R3-W1: Methodological  contribution**
>
> **Response:**  We respectfully clarify this point. Our primary novelty lies in pioneering a highly valuable, underexplored problem: efficient dense-to-MoE conversion for visual autoregressive generation. While we build upon established concepts like pruning or routing, tackling this challenge is strictly harder than standard upcycling in language models. A naive combination falls short due to the unique spatial dependencies of visual tokens, necessitating our synergistic framework. In this sense, our contribution lies in formulating and addressing this specific problem setting, together with a targeted method design including trajectory-consistent initialization and confidence-adaptive sparse fine-tuning.
>
> ---
>
> **R3-W2:**
>
> **Response:** We additionally evaluated Prism-MoE on both **a larger backbone** and **a different backbone** .
>
> **Table 1. Larger-scale evaluation on Infinity-8B.**
>
> | **Activated params** | **GenEval ↑** | **DPG-Bench ↑** |
> | --- | --- | --- |
> | Dense | 0.80 | 86.6 |
> | 75% active, w/o ft | 0.78 | 86.4 |
>
> **Table 2. Results of Prism-MoE on Semanticist.**
>
> | **Model** |  **FID** |
> | --- | --- |
> | Dense | 2.49 |
> | Prism-MoE, 75% active, w/o ft |  2.99 |
> | Prism-MoE, 37.5% active, w/ ft | 2.50 |
>
> These additional results provide complementary evidence: on Infinity-8B, Prism-MoE remains close to the dense baseline, and on Semanticist, it also transfers beyond the backbones.
>
> ---
>
> **R3-W3:  efficiency analysis**
>
> **Response:** To complement generation quality and parameter efficiency, we additionally report system-level metrics, including latency, throughput, and memory usage.
>
> **Table 3. End-to-end inference efficiency on VAR-d20**
>
> | **Model** | **Latency** | **Throughput** | **Memory** |
> | --- | --- | --- | --- |
> | Dense | 45.5 ms | 22.0 it/s | 79293 MB |
> | Prism-MoE, 37.5% active | 37.2 ms | 26.9 it/s | 50879 MB |
>
> **Table 4. End-to-end inference efficiency on LlamaGen-XL Stage I**
>
> | **Model** | **Latency** | **Throughput** | **Memory** |
> | --- | --- | --- | --- |
> | Dense | 293.6 ms | 3.41 it/s | 60953 MB |
> | Prism-MoE, 37.5% active | 237.6 ms | 4.21 it/s | 46497 MB |
>
> These results show that Prism-MoE improves not only parameter efficiency, but also practical inference efficiency.
>
> ---
>
> **R3-W4: Discussion and generalizability**
>
> **Response:**  We would like to clarify this point. The paper already motivates why dense-to-MoE conversion is especially challenging in visual autoregressive generation, due to step-wise trajectory non-stationarity, causal dependency, and irreversible error accumulation. While our current scope is visual autoregressive generation, the additional result in Table 2 also shows that Prism-MoE transfers to Semanticist.
>
> ---
>
> **R3-Q1: Scalability to larger models**
>
> **Response:** Please see Response to R3-W1, Table 1. The added result on Infinity-8B shows that Prism-MoE remains close to the dense baseline on a multi-billion-parameter backbone.
>
> ---
>
> **R3-Q2: Inference latency**
>
> **Response:** Please see Response to R3-W3, Tables 3 and 4. We added system-level measurements including latency, throughput, and memory usage.
>
> ---
>
> **R3-Q3: Sensitivity to initialization**
>
> **Response:** We examined the sensitivity to the shared versus routed partition ratio while keeping the overall sparse setting fixed.
>
> **Table 5. Sensitivity to the shared expert ratio**
>
> | **n_experts** | **shared ratio** | **activated ratio** | **FID** |
> | --- | --- | --- | --- |
> | 16 | 0.25 | 0.125 | 2.86 |
> | 16 | 0.125 | 0.25 | 2.91 |
>
> These results show that performance remains stable across nearby partition choices. Our final setting follows the shared-expert design in Dense2MoE [1], which also reports that making the shared expert too small can hurt optimization and overall performance. This is consistent with our observation.
>
>
> [1] Dense2moe: Restructuring diffusion transformer to moe for efficient text-to-image generation. ICCV, 2025.
>
>
> ---
>
> **R3-Q4: Generalization**
>
> **Response:**  Our method is designed for visual autoregressive generation. Please see Response to R3-W2, Table 2 for the added result on Semanticist on broader backbones.
>
> ---
>
> **R3-L1: Technical limitations**
>
> **Response:** We would like to clarify that the paper focuses on visual autoregressive generation, covering both next-scale (VAR) and next-token (LlamaGen) paradigms. In the rebuttal, we further added a larger-scale result on Infinity-8B, while the sensitivity analysis is supported by Response to R1-W2, Table 3.
>
> ---
>
> **R3-L2: Practical deployment**
>
> **Response:** Please see Response to R3-W3, where we added system-level measurements.
>
>
> ---
>
> **R3-L3: Broader impact discussion**
>
> **Response:** Our work is primarily focused on efficiency rather than downstream use cases. We will add a brief discussion of this aspect in the revised broader impact section.

---

> > ### Author Rebuttal · Reviewer_Y9cC · 2026-04-04
> >
> > I appreciate the authors’ clarifications and additional experiments, which largely address my concerns, and I agree that the proposed method is both interesting and practically useful, so I am inclined to raise my score.

---

> > > ### Author Response · Authors · 2026-04-04
> > >
> > > Dear Reviewer Y9cC,
> > >
> > > We sincerely appreciate your thoughtful evaluation of our work and your decision to raise the score. We are particularly encouraged by your recognition that the paper **addresses a relevant and timely problem**, and by your positive assessment of Prism-MoE as a **practical** dense-to-MoE conversion strategy. We also value your comments that the two-stage framework is **logically motivated**, that the empirical evaluation generally supports the method’s effectiveness, and that the paper is **clearly written, well structured, and coherent in its motivation, design, and validation**.
> > >
> > > We are glad that the additional experiments and clarifications helped address your concerns, and that they strengthened your view that the method is both **interesting and practically useful**. Your feedback has been very helpful in improving the paper.
> > >
> > > Thank you again for your time, expertise, and constructive feedback.
> > >
> > > Best regards,
> > > Authors

---

### Official Review · Reviewer_KLTj · 2026-03-12

**Soundness:** 3
**Presentation:** 3
**Significance:** 3
**Originality:** 2
**Overall Recommendation:** 5
**Confidence:** 3

**Summary:**

In this work, the authors propose Prism-MoE, a two-stage framework that can be used to convert autoregressive models for image generation into sparse MoE versions of them. On the first stage, they perform trajectory expert initialization, while on the second stage, the authors introduce confidence-adaptive fine tuning with a normalized cosint router. On the experimental section, the authors show that the method reaches roughly the same image generation quality, while having a significantly lower number of active parameters.

**Compliance With Llm Reviewing Policy:**

Affirmed.

**Final Justification:**

I thank the reviewers for a strong rebuttal, addressing my concerns.

My main points were on the number of parameters (which was a misunderstanding from my part), ablations on d16 and d20 (which the authors provided), extending the work to AR-diffusion models (which the authors provided preliminary experiments), training from scratch and also the wall-clock latency.

The authors showed numbers on all of these experiments, strengthening their paper and my conviction for it. The authors also provided experiments (asked by another reviewer) on a larger backbone.

Considering all of these, I am switching my score from Weak Reject (3) to Accept (5), and now I support this paper to be published.

**Key Questions For Authors:**

1) Is the 37.5%, the number of parameters compared to the MLP, or compared to the total number of parameters?

2) Does the method work with hybrid AR-diffusion generative models such as Semanticist [A]. I guess the authors can explore and potentially discuss this.

[A] Wen et al.,  Principal Components Enable A New Language of Images, ICCV 2025

3) Why the ablation results were provided on d16 instead of d20?

4) Can the authors provide a more details analysin on what the experts actually learn under this framework? Do they specialize on different spatial frequencies, different object classes, different generation scales etc?

I might increase my score if the authors provide clear answers in the points above.

**Limitations:**

I think the paper can be improved on the following aspect:

1) There is no comparisons against training a MoE model from scratch, or against training the dense model for the same number of additional epochs. Without this, it is a bit unclear if the proposes framework is better than other alternatives.

2) While I appreciate the evaluation on the number of FLOPs, I think the authors should have provided also the wall-clock latency numbers which can introduce some inference overhead (routing, expert balancing, etc.)

3) There is a bit of missmatch between the main results, provided on d20, and the ablation results, provided on d16? It is slightly unclear to me if the ablation results would scale to d20.

4) It would be nice if the authors can provide a more detailed analysis on expert specialization (see question 4).

**Strengths And Weaknesses:**

I think the paper has the following strengths:

1) The paper is well-motivated. Mixture of experts (MoE) have played a significant role in the modern LLMs, to the point that almost every good open-source LLM has been replacing dense MLPs with mixture of experts MoE. It is logical, that the same can happen in AR models for image generation.

2) The paper shows .quite good results on VAR-d20. They basically match VAR on FID score, while using roughly 1/3rd of the active parameters. Also, the ablations on Table 3 are quite sensible.

3) The two-stage approach makes quite a bit sense to me, starting from a training-free initialization and then following with lightweight fine-tuning. Furthermore, the overall computational overhead (~10% of the training budget) makes the method could useful in practice.

---

> ### Author Rebuttal · Authors · 2026-03-31
>
> We thank the reviewer for the positive assessment and valuable suggestions.
>
> **R2-Q1: “activation ratio...”**
>
> **Response:** It refers to the **FFN/MLP activation ratio**, not the total model parameters.
>
> ---
>
> **R2-Q2: “Does the method work with hybrid AR-diffusion generative models …”**
>
> **Response:**  We tested Prism-MoE on Semanticist with a DiT-L backbone, representing a hybrid AR-diffusion setting.
>
> **Table 1. Results of Prism-MoE on Semanticist.**
>
> | **Model** | **FID** |
> | --- | --- |
> | Dense | 2.49 |
> | Prism-MoE, 75% act., w/o ft  | 2.99 |
> | Prism-MoE, 37.5% act., w/ ft | 2.50 |
>
> These preliminary results suggest that the method can also transfer beyond pure AR visual generation.  We will explore this direction for further study.
>
> ---
>
> **R2-Q3: “ablation results on d20”**
>
> **Response:** We used VAR-d16 for ablation because it shares the same architecture and C2I setting as VAR-d20, while being more efficient for controlled component analysis. To address the reviewer’s concern, we additionally ran the same ablation on VAR-d20, and Table 2 shows the same trend.
>
> **Table 2. Ablation results on VAR-d20.**
>
> | **w/ FT** | **Configuration** | **Act.** | **FID** |
> | --- | --- | --- | --- |
> | N | Init  | 75% | 15.0 |
> | N | + Trajectory-Consistent Init | 75% | 5.03 |
> | Y | + Fine-tuning (Distillation only) | 37.5% | 10.1 |
> | Y | + Normalized Cosine Router | 37.5% | 3.55 |
> | Y | + Spatial Loss (Full Prism-MoE) | 37.5% | 2.86 |
>
> ---
>
> **R2-Q4: “experts behavior”**
>
> **Response:** We further analyzed what the experts learn under Prism-MoE from two perspectives: generation stage specialization and uncertainty-aware specialization. Table 3 reports representative layer-specific experts with the largest high versus low entropy activation differences, and we also provide a supplementary visualization of expert specialization across generation stages and uncertainty levels here:
> (https://anonymous.4open.science/r/Prism-MoE-rebuttal-material-CC23/Expert%20specialization%20across%20generation%20stages%20and%20uncertainty%20levels..jpeg)
>
> Table 3 and the supplementary figure both show layer-specific expert specialization, indicating that Prism-MoE learns non-uniform, differentiated routing rather than homogeneous sparse routing.
>
> **Table 3. Layer-specific expert responses to token uncertainty.**
>
> | **Layer** | **Most entropy sensitive expert** | **Second most entropy sensitive expert** |
> | --- | --- | --- |
> | Layer 10 | Expert 0, **+11.15 pp** | Expert 5, **-8.44 pp** |
> | Layer 12 | Expert 11, **-10.73 pp** | Expert 0, **+6.65 pp** |
> | Layer 14 | Expert 8, **-9.91 pp** | Expert 0, **+8.07 pp** |
> | Layer 16 | Expert 1, **+8.49 pp** | Expert 4, **-7.32 pp** |
>
> ---
>
> **R2-L1: “comparisons against training from scratch…”**
>
> **Response:** We additionally compared against training the MoE model from scratch on VAR-d20 under the same 20 epoch budget. The results show that Prism-MoE is clearly more effective under the same limited training budget.
>
> **Table 4. Comparison with MoE training from scratch on VAR-d20**
>
> | **Method** | **FID** |
> | --- | --- |
> | MoE trained from scratch, 20 epochs | 8.99 |
> | Prism-MoE initialization, w/o ft | 5.03 |
> | Prism-MoE, w/ ft for 20 epochs | 2.86 |
>
> Dense training serves a different purpose: it does not provide the sparse activation capability targeted by our method. Due to rebuttal time and resource limits, we were unable to include this experiment.
>
> ---
>
> **R2-L2: “wall-clock latency…”**
>
> **Response:** We added inference measurements on A100 80G, batch size 128.
>
> **Table 5. VAR-d20**
>
> | **Model** | **Latency** | **Throughput** | **Memory** |
> | --- | --- | --- | --- |
> | Dense | 45.5 ms | 22.0 it/s | 79293 MB |
> | Prism-MoE, 37.5% active | 37.2 ms | 26.9 it/s | 50879 MB |
>
> **Table 6. LlamaGen-XL Stage I**
>
> | **Model** | **Latency** | **Throughput** | **Memory** |
> | --- | --- | --- | --- |
> | Dense | 293.6 ms | 3.41 it/s | 60953 MB |
> | Prism-MoE, 37.5% active | 237.6 ms | 4.21 it/s | 48497 MB |
>
> **Table 7. Wall clock time breakdown for VAR-d20 **
>
> | **Component** | **Ratio** |
> | --- | --- |
> | Core generation (infer_moe) | 55.24% |
> | Expert statistics collection | 6.27% |
> | PNG saving | 38.49% |
>
> These results complement FLOPs with direct system measurements. On both VAR-d20 and LlamaGen-XL Stage I, the 37.5% activation model improves latency, throughput, and memory usage over the dense baseline. Moreover, Table 7 shows that the additional expert-related overhead is limited.
>
> ---
>
> **R2-L3: “ablations on d16…”**
>
> **Response:**  Please see our **Response to R2-Q3**, where we additionally provide the VAR-d20 ablation.
>
> ---
>
> **R2-L4: “expert specialization…”**
>
> **Response:**  Please see our Response to R2-Q4, which shows that Prism-MoE learns layer-specific expert specialization rather than uniform sparse routing.
>
> We are **actively available** during the next Author-Reviewer Discussion period. Please let us know if you have any further questions regarding our responses. Thanks!

---

> > ### Author Rebuttal · Reviewer_KLTj · 2026-04-03
> >
> > I thank the reviewers for a strong rebuttal, addressing my concerns.
> >
> > My main points were on the number of parameters (which was a misunderstanding from my part), ablations on d16 and d20 (which the authors provided), extending the work to AR-diffusion models (which the authors provided preliminary experiments), training from scratch and also the wall-clock latency.
> >
> > The authors showed numbers on all of these experiments, strengthening their paper and my conviction for it. The authors also provided experiments (asked by another reviewer) on a larger backbone.
> >
> > Considering all of these, I am switching my score from Weak Reject (3) to Accept (5), and now I support this paper to be published.

---

> > > ### Author Response · Authors · 2026-04-03
> > >
> > > Dear Reviewer KLTj
> > >
> > > We sincerely appreciate your thoughtful evaluation of our work and your decision to raise the score. Your feedback was highly valuable in helping us strengthen the paper.  We are especially encouraged by your recognition of the paper’s motivation, the practicality of the two-stage design, and the strong efficiency-quality tradeoff achieved by Prism-MoE.
> > >
> > > We are grateful that you found the additional experiments and clarifications helpful, and that they strengthened your confidence in the work. It has been a privilege to engage with your review, which has materially improved the final version of the paper.
> > >
> > > Thank you again for your time, expertise, and constructive feedback.
> > >
> > > Best regards,
> > > Authors

---

### Official Review · Reviewer_rhJx · 2026-03-12

**Soundness:** 3
**Presentation:** 3
**Significance:** 3
**Originality:** 3
**Overall Recommendation:** 4
**Confidence:** 4

**Summary:**

This paper proposes Prism-MoE, a two-stage framework for converting pretrained dense visual autoregressive generators (next-scale VAR and next-token LlamaGen) into sparse Mixture-of-Experts (MoE) models with minimal additional training. Stage I introduces trajectory-consistent expert initialization that partitions neurons using importance and activation stability and then applies a closed-form, ridge-based residual compensation to match the dense model’s step-wise generation trajectory. Stage II performs condition-adaptive sparse fine-tuning via a magnitude-invariant cosine router with context bias and spatially structured routing regularization weighted by token-level uncertainty. Across class-to-image and text-to-image settings, the method achieves near-dense quality with only 37.5% active parameters using less than 10% of dense training budget.

**Compliance With Llm Reviewing Policy:**

Affirmed.

**Final Justification:**

Thanks for the authors' detailed rebuttal. I admire the authors work a lot and I will keep score.

**Key Questions For Authors:**

1. How sensitive is Prism-MoE to the calibration set and decomposition hyperparameters in Stage I?
2. What is the actual end-to-end latency improvement, not just FLOPs reduction, under realistic inference settings?
3. How does the method behave under more aggressive sparsity or on larger backbones, such as infinity-8B?

**Limitations:**

Yes

**Strengths And Weaknesses:**

Strengths:
1. Clear motivation: The paper addresses a practically relevant problem: how to obtain the inference efficiency benefits of MoE for visual autoregressive models without incurring the prohibitive cost of training MoE models from scratch.
2. Good problem framing: The paper also clearly identifies why this setting is harder than standard MoE upcycling in language or diffusion: autoregressive generation has strong causal dependencies, step-wise non-stationarity, and irreversible error accumulation.
3. Experimental rigor and validation: The paper evaluates both next-scale (VAR-d16/d20 on ImageNet-1K) and next-token (LlamaGen-XL Stage I on COCO prompts) paradigms, demonstrating cross-paradigm applicability. Careful ablations isolate the contribution of each component (trajectory-consistent init, cosine router, spatial loss), showing consistent, additive gains.

Weaknesses:
1. Experimental gaps: Efficiency reporting focuses on FLOPs/active parameters; there is no end-to-end latency, throughput, or memory profiling under realistic distributed MoE inference (e.g., all-to-all costs), which is critical for deployment claims.
2. Technical concerns: The method’s success hinges on a calibration procedure (collecting hidden states and outputs for ridge correction), but sensitivity to calibration set size/content and robustness across domains are not quantified. The text-to-image evaluation uses 5k COCO validation prompts; broader T2I metrics (e.g., GenEval, HPSv2, DPG-Bench) would strengthen claims.

---

> ### Author Rebuttal · Authors · 2026-03-31
>
> We thank the reviewer for the valuable feedback, and we respond to each point below.
>
> **R1-W1: End-to-end costs**
>
> **Response:**  Following this suggestion, we added direct inference measurements on **A100 80G, batch size 128**, using the same evaluation setting as in the main paper.
>
> **Table 1. End-to-end inference efficiency on VAR-d20.**
>
> | **Model** | **Latency** | **Throughput** | **Memory** |
> | --- | --- | --- | --- |
> | Dense | 45.5 ms | 22.0 it/s | 79293 MB |
> | Prism-MoE, 37.5% active | 37.2 ms | 26.9 it/s | 50879 MB |
>
> **Table 2. End-to-end inference efficiency on LlamaGen-XL Stage I.**
>
> | **Model** | **Latency** | **Throughput** | **Memory** |
> | --- | --- | --- | --- |
> | Dense | 293.6 ms | 3.41 it/s | 60953 MB |
> | Prism-MoE, 37.5% active | 237.6 ms | 4.21 it/s | 46497 MB |
>
> These results complement FLOPs with direct system measurements. On both VAR-d20 and LlamaGen-XL Stage I, the 37.5% active Prism-MoE model improves latency, throughput, and memory over the dense baseline. Our current measurements are obtained under single-GPU inference, so they do not include expert-parallel communication overhead such as all-to-all. We will clarify this scope in the revision.
>
> ---
>
> **R1-W2: Calibration sensitivity and T2I evaluations**
>
> **Response:** We added additional experiments on **calibration set size**, **ridge regularization sensitivity**, and **broader text-to-image benchmarks**.
>
> **Table 3. Sensitivity to calibration set size.**
>
> | **Calibration set size** | **VAR-d20 FID** | **LlamaGen-XL Stage I FID** |
> | --- | --- | --- |
> | 100 | 6.04 | 27.4 |
> | 200 | 5.03 | 27.1 |
> | 300 | 4.96 | 27.0 |
>
> These results show that the method is **not overly sensitive** to the calibration set size. Increasing the calibration set from **100 to 300** yields only modest improvements.
>
> **Table 4. Sensitivity to the adaptive ridge multiplier.**
>
> | **Adaptive ridge multiplier** $\alpha$ | **VAR-d20 FID** | **LlamaGen-XL Stage I FID**  |
> | --- | --- | --- |
> | 0.5 | 5.06 | 27.0 |
> | 1.0 | 5.03 | 27.1 |
> | 2.0 | 5.28 | 27.3 |
>
> These results indicate that the initialization is also **reasonably robust** to the ridge regularization strength.
>
> To strengthen the T2I evaluation beyond the **5K COCO validation prompts**, we further evaluated **LlamaGen-XL Stage I** on additional benchmarks.
>
> **Table 5. Additional text-to-image evaluation on broader benchmarks.**
>
> | **Model** | **GenEval** $\uparrow$ | **HPSv2** $\uparrow$ |
> | --- | --- | --- |
> | Dense | 0.264 | 18.65 |
> | Prism-MoE, 75% active, w/o ft | 0.235 | 14.25 |
> | Prism-MoE, 37.5% active, w/ ft | 0.271 | 18.71 |
>
> These results further support the effectiveness of Prism-MoE in the T2I setting. Overall, the additional experiments suggest that Prism-MoE is reasonably robust to calibration choices and remains effective on broader T2I benchmarks.
>
> ---
>
> **R1-Q1: Calibration sensitivity**
>
> **Response:** Please see our **Response to R1-W2**. The additional results show that the Stage I calibration and decomposition procedure remains stable across different calibration set sizes and adaptive ridge multipliers.
>
> ---
>
> **R1-Q2: end-to-end latency**
>
> **Response:**  Please see our **Response to R1-W1**. The added wall-clock measurements show clear end-to-end efficiency gains under realistic single-GPU inference on both VAR-d20 and LlamaGen-XL Stage I.
>
> ---
>
> **R1-Q3: aggressive sparsity and larger backbones**
>
> **Response:**  We thank the reviewer for this suggestion and added additional results on both **more aggressive sparsity** and a **larger backbone**.
>
> **Table 6. More aggressive sparsity on VAR-d20.**
>
> | **Activated params** | **FID** |
> | --- | --- |
> | 62.5%, w/o fine-tuning | 13.65 |
> | 31.25%, w/ fine-tuning | 4.53 |
>
> These results suggest that Prism-MoE remains viable under more aggressive sparsity, although some degradation is expected compared with the main setting. Still, the framework preserves reasonable generation quality even in a stricter sparse regime.
>
> **Table 7. Larger-backbone evaluation on Infinity-8B.**
>
> | **Activated params** | **GenEval** $\uparrow$ | **DPG-Bench** $\uparrow$ |
> | --- | --- | --- |
> | Dense | 0.80 | 86.6 |
> | 75%, w/o fine-tuning | 0.78 | 86.4 |
>
> These preliminary results suggest that Prism-MoE also transfers well to a **larger backbone**. On **Infinity-8B**, the **75% active** model without fine-tuning remains very close to the dense baseline on both **GenEval** and **DPG-Bench**, indicating that the proposed initialization still scales reasonably well in the large-model setting.
>
> For the fine-tuned Infinity-8B result, we agree that it would be valuable. However, fine-tuning for 20 epochs requires more than 30 days on 8×A100. Due to the limited rebuttal time and compute budget, we were unable to include this experiment.

---

> > ### Author Rebuttal · Reviewer_rhJx · 2026-04-07
> >
> > Thank you so much for the detailed response.

---

> > > ### Author Response · Authors · 2026-04-08
> > >
> > > Dear Reviewer rhJx,
> > >
> > > Thank you for your thoughtful evaluation of our work and for carefully considering our rebuttal. We greatly appreciate your recognition of the paper’s **clear motivation, good problem framing,** and **experimental rigor and validation.** We are particularly encouraged that you highlighted the problem as practically relevant and recognized the specific challenges of visual autoregressive generation, including **strong causal dependencies, step-wise non-stationarity,** and irreversible error accumulation.
> > >
> > > Your feedback has been very helpful in improving the paper and sharpening our presentation. We are grateful for your time, expertise, and constructive review.
> > >
> > > Best regards,
> > > Authors

---

### Decision · Program_Chairs · 2026-04-30

**Decision:**

Accept (regular)

**Comment:**

The paper proposes Prism-MoE, a two-stage framework for converting pretrained dense visual autoregressive generators into sparse Mixture-of-Experts models without full retraining. The key idea is to first partitions neurons using importance and activation statistics, and then apply a confidence-aware sparse fine-tuning stage with spatial regularization, and expert-diversity constraints.
During the discussion phase, reviewers raised concerns regarding wall-clock latency, generalization to other backbones and architectures, and the comprehensiveness of the ablation studies. The rebuttal effectively addresses most of these points: the authors present supplementary latency and memory measurements, incorporate calibration sensitivity analyses, and report additional experimental results on the Infinity model. Concerning novelty and broader impact, the authors also provide a well-reasoned defense of the method’s technical validity and real-world applicability.
Overall, the paper represents a solid contribution to the research community, and the AC recommends acceptance.